# Should Procalcitonin Be Included in Acute Cholecystitis Guidelines? A Systematic Review

**DOI:** 10.3390/medicina59040805

**Published:** 2023-04-20

**Authors:** Clyve Yu Leon Yaow, Ryan Ian Houe Chong, Kai Siang Chan, Christopher Tze Wei Chia, Vishal G. Shelat

**Affiliations:** 1Yong Loo Lin School of Medicine, National University of Singapore, Singapore 117597, Singapore; e0268630@u.nus.edu (C.Y.L.Y.); e0491492@u.nus.edu (R.I.H.C.); 2Department of General Surgery, Tan Tock Seng Hospital, Singapore 637551, Singapore; kchan023@e.ntu.edu.sg; 3Department of Gastroenterology and Hepatology, Tan Tock Seng Hospital, Singapore 637551, Singapore; 4Lee Kong Chian School of Medicine, Nanyang Technological University, Singapore 308232, Singapore

**Keywords:** acute cholecystitis, gallstones, inflammatory markers, procalcitonin, prognosis

## Abstract

*Background and Objectives*: Acute cholecystitis (AC) is a common surgical emergency. Recent evidence suggests that serum procalcitonin (PCT) is superior to leukocytosis and serum C-reactive protein in the diagnosis and severity stratification of acute infections. This review evaluates the role of PCT in AC diagnosis, severity stratification, and management. *Materials and Methods*: PubMed, Embase, and Scopus were searched from inception till 21 August 2022 for studies reporting the role of PCT in AC. A qualitative analysis of the existing literature was conducted. *Results*: Five articles, including 688 patients, were included. PCT ≤ 0.52 ng/mL had fair discriminative ability (Area under the curve (AUC) 0.721, *p* < 0.001) to differentiate Grade 1 from Grade 2–3 AC, and PCT > 0.8 ng/mL had good discriminatory ability to differentiate Grade 3 from 1–2 AC (AUC 0.813, *p* < 0.001). PCT cut-off ≥ 1.50 ng/mL predicted difficult laparoscopic cholecystectomy (sensitivity 91.3%, specificity 76.8%). The incidence of open conversion was higher with PCT ≥ 1 ng/mL (32.4% vs. 14.6%, *p* = 0.013). A PCT value of >0.09 ng/mL could predict major complications (defined as open conversion, mechanical ventilation, and death). *Conclusions*: Current evidence is plagued by the heterogeneity of small sample studies. Though PCT has some role in assessing severity and predicting difficult cholecystectomy, and postoperative complications in AC patients, more evidence is necessary to validate its use.

## 1. Introduction

Acute cholecystitis (AC) is a common surgical admission and an important cause of morbidity and mortality [1,2]. The lifetime prevalence of a healthy adult having gallstones is estimated to be about 15–20%, with a wide geographic variation, with 20% of these patients developing AC [3]. More than 200,000 patients are diagnosed with AC in the USA annually [4,5,6,7]. Gallstones with resultant cystic duct occlusion and increased intraluminal pressure account for most AC episodes [8,9]. Though AC is defined as sepsis, microbial isolation is not uniformly reported, and inflammatory markers are included in diagnostic criteria [10].

In routine practice, clinicians use raised serum white blood cell count (WBC), i.e., leukocytosis and C-reactive protein (CRP) levels to aid diagnosis, stratify severity and predict clinical progress and operative difficulty in AC management [11,12,13,14,15]. However, these markers have limitations. For example, leukocytosis is a non-specific marker for sepsis and WBC count may be paradoxically reduced in older, diabetic, immunosuppressed patients, and in patients with severe sepsis [16]. A retrospective study by Yazici et al. [17] found that 43% of AC patients (n = 31/72) had normal WBC count. Furthermore, Hwang et al. [18] retrospectively studied 107 patients managed by emergency cholecystectomy and reported that a WBC count of >9.7 × 10^9^/L has only 64% sensitivity and 47% specificity for diagnosing AC. CRP has been shown to have better discriminative power in AC diagnosis than WBC [19]. However, the elevation of CRP is reported in only 55.1% to 65.3% of patients [17,20]. Yazici et al. [17] found that 34.7% of AC patients (n = 25/72) had normal CRP levels. The definition of elevated CRP for severity stratification of AC varies across studies [16,21]. The use of CRP is also limited in patients with acute fulminant liver failure and patients on medications such as corticosteroids and hematological therapies [22,23,24].

Procalcitonin (PCT) is an increasingly used biomarker that is used to differentiate infections from non-infective causes of inflammation, with higher sensitivity (88% vs. 75%) and specificity (81% vs. 67%) than CRP [1,25,26,27,28,29,30,31,32]. At the time of the Tokyo Guidelines 2018 (TG18) revision, there was limited evidence on the utility of PCT in AC care. Thus, it was not included within the guidelines. Since then, there have been studies evaluating the role of PCT in diagnosis, severity stratification, and/or outcomes in AC [33,34]. It is timely that a comprehensive review of existing evidence is conducted to shed light if PCT could be included in AC management. To the best of our knowledge, there is no review summarizing the utility of PCT in AC. Hence, this systematic review aims to evaluate the role of PCT in the diagnosis, severity stratification, and management of AC.

## 2. Materials and Methods

### 2.1. Search Strategy and Selection Criteria

This review protocol was guided by the latest Preferred Reporting Items for Systematic Reviews and Meta-Analyses (PRISMA) guidelines [35] and registered on PROSPERO (CRD42022342705). PubMed, Embase, Scopus, Web of Science, and the Cochrane Library were searched from inception till 21 August 2022 for articles studying the role of PCT in AC. The search was restricted to titles and abstracts for all databases. The following search terms were used in combination: (“procalcitonin” OR “PCT”) AND (“cholecystitis” AND “acute cholecystitis”). The detailed search strategy is appended in Appendix A. The PRISMA checklist is appended in Appendix A. A manual search of references cited in the final included articles was also performed to identify other reports.

Inclusion criteria were randomized controlled trials (RCTs) or cohort studies comparing PCT with WBC and/or CRP in diagnosis, severity stratification, and management of AC. Exclusion criteria were (1) articles that evaluated other inflammatory markers (e.g., raised WBC count and CRP) without evaluation of PCT, (2) hepatobiliary infections (e.g., acute cholangitis) other than AC, (3) article type (case series, case reports, reviews, opinions, editorials and conference abstracts), and (4) non-English articles. Any disagreements were resolved by discussion and consensus with the senior author (VGS).

After the removal of duplicates, two independent authors (CYLY, RIHC) screened the title and abstracts for potential inclusion using the above-defined inclusion and exclusion criteria. Full texts were subsequently reviewed in their entirety for eligibility in the final review. Conflicts were resolved by discussion with the senior author (VGS). This process is illustrated in the PRISMA flow diagram (Figure 1).

### 2.2. Data Extraction and Synthesis

Key information about the articles was extracted into a predefined datasheet by two independent authors (CYLY, RIHC). Study characteristics include author, year of publication, country, study period, patient characteristics (age and comorbidities), laboratory values (PCT levels, WBC levels, CRP levels), method of diagnosis and severity stratification of AC, intra-operative complications (operative time, blood loss, conversion to open, and difficult laparoscopic cholecystectomy (DLC)) and postoperative complications (major complications, progression to sepsis, length of stay (LOS), and mortality). Tokyo Guidelines 2013 (TG13) and Tokyo Guidelines 2018 (TG18) diagnostic criteria define severe AC (Grade 3) when a patient has any dysfunction of cardiovascular, neurological, respiratory, renal, hepatic, or hematological systems and moderate AC (Grade 2) when a patient has symptom duration of >72 h, palpable tender right upper quadrant mass, elevated WBC > 18,000/mm^3^, or imaging features of marked local inflammation such as gangrene, pericholecystic abscess, hepatic abscess, biliary peritonitis, or emphysematous cholecystitis [1]. Wu et al. defined DLC as a total operation time of more than 120 min, more than 40 min to obtain a critical view of safety or conversion to open surgery due to technical difficulties or complications [36]. Major complications were defined by Fransvea et al. [33] as death, the need for mechanical ventilation, and conversion to open surgery. Fransvea et al. [33] defined progression to sepsis based on clinical suspicion of infection and quick Sequential Organ Failure Assessment (qSOFA) score ≥ 2. Mortality was defined as 30-day mortality (i.e., death within 30 days from admission) [33]. Any discrepancies were resolved by consensus and discussed with a co-author (KSC). Outcomes were synthesized with a narrative synthesis and tabular presentation of quantitative data. Summary tables describing article characteristics and findings were also presented. Continuous variables were described as mean ± standard deviation unless otherwise specified. Quantitative data using mean and standard deviation were pooled together using methods described by Altman et al. [37]. A meta-analysis was not performed given the heterogeneity included studies.

### 2.3. Risk of Bias Assessment

Two independent authors (CYLY, RIHC) performed quality assessments for observational studies using the modified Newcastle-Ottawa Scale (Appendix A) [38]. Disagreements between authors were resolved by discussion with a co-author (KSC).

## 3. Results

### 3.1. Study Characteristics

The initial search yielded 4845 records, from which 1046 duplicates were removed. Of the remaining 3799 records, 3792 records were excluded based on their titles and abstracts as they did not study PCT levels or patients with AC (Figure 1).

Seven full texts were reviewed, of which four articles were eligible [25,33,34,39]. One additional article was identified from reviewing references of eligible articles [36] Therefore, the final analysis included five articles (four prospective cohort studies and one retrospective study). Study characteristics and patient demographics are summarized in Table 1.

### 3.2. Patient Demographics

There were 688 AC patients with an overall mean age of 59.59 ± 16.25 years, and 46.4% were male. Two studies used TG13 while three studies used TG18 for diagnosis and severity stratification of AC. Significant comorbidities, such as diabetes mellitus and peripheral vascular disease, were reported only by Fransvea et al. and Wu et al. [33,36]. Other patient characteristics are summarized in Table 1. The pooled overall mean PCT level was 0.78 ± 1.02 ng/mL. Four authors reported CRP values, and the mean CRP was 29.58 ± 45.17 ng/mL [25,33,34,36]. Subgroup analysis was performed for included patients in their respective studies based on PCT levels on admission, grade of AC, and difficulty of laparoscopic cholecystectomy [25,33,34,36,39].

### 3.3. Severity of AC

Three studies (n = 399) reported correlations between PCT and the severity of AC (Table 2) [25,34,39]. Evidence on whether PCT could discriminate between grades of AC was equivocal. Three patterns of PCT utility in AC severity stratification are evident. Firstly, the PCT level was correlated with the severity of AC using Spearman’s correlation [34]. Secondly, PCT levels at various cut-off values could discriminate Grade 1 from Grade 2–3 severity [25]. Thirdly, PCT levels were similarly able to distinguish Grade 3 from Grade 1–2 severity [25]. Naz et al. observed that PCT values were raised in 13.79% of Grade 1, 40.63% of Grade 2, and 78.57% of Grade 3 AC patients [39]. However, the authors did not define a cut-off value for “raised PCT”. Two studies reported on the discriminatory ability of PCT between various severity Grades of AC. Yuzbasioglu et al. observed that PCT ≤ 0.52 ng/mL had fair discriminative ability (Area under the curve (AUC) 0.721, *p* < 0.001) to differentiate Grade 1 from Grade 2–3 AC with 95.45% sensitivity and 46.67% [25]. However, Naz et al. observed that PCT values could not differentiate Grade 1 from Grade 2–3 AC, with 91.7% sensitivity and 20.6% specificity, at a cut-off value of 0.15 ng/mL (AUC 0.439, *p* = 0.43) [39]. Results were similar when investigating whether PCT was able to differentiate Grade 3 AC from Grade 1–2 AC: Yuzbasioglu et al. found that PCT > 0.8 ng/mL had a good discriminatory ability to differentiate Grade 3 from 1–2 AC (AUC 0.813, *p* < 0.001) [25], but not in the study by Naz et al. (cut-off > 11.5 ng/mL, *p* = 0.535) [39].

### 3.4. Intra-Operative Complications

Wu et al. reported the ability of PCT to predict DLC (Odds ratio (OR): 408.52, 95% CI: 41.17–4053.27, *p* < 0.001). The PCT cut-off of 1.50 ng/mL with the sensitivity and specificity of 91.3% (95% CI 78.3–97.1) and 76.8% (95% CI 64.8–85.8) was reported. Patients with PCT ≥ 1.50 ng/mL had a significantly higher incidence of DLC than those with PCT < 1.50 ng/mL (OR 5.2, 95% CI: 3.7–7.5, *p* = 0.004) [36].

Conversion from laparoscopic to open cholecystectomy surgery was reported by two authors (11.42%, n = 33/289) [33,36]. Fransvea et al. reported a higher risk of open conversion in patients with PCT ≥1 ng/mL compared to <1 ng/mL (32.4% vs. 14.6%, *p* = 0.013) [33]. On the other hand, Wu et al. found no significant difference in conversion rates in DLC and non-DLC (NDLC) (*p* = 0.4) [36].

Other intra-operative complications, including operative time, dense adhesions, blood loss, and perforation of the gallbladder, were reported in 115 patients [36]. Between DLC and NDLC, a significant difference in operative time (DLC 147.4 ± 17.9 min vs. NDLC 66.8 ± 13.4 min, *p* < 0.001), presence of dense adhesions (DLC n = 20 vs. NDLC n = 15, *p* = 0.013) and blood loss (DLC 70.9 ± 33.9 mL vs. DLC 15.9 ± 9.9 mL, *p* < 0.001) was reported [36].

### 3.5. Post-Operative Complications

Wu et al. reported a higher incidence of Clavien-Dindo Grade 1–2 complications in DLC compared to NDLC (15.2% vs. 2.9%, *p* < 0.001) [36]. Fransvea et al. reported the overall incidence of major complications was 19% (n = 33/174) [33]. Multivariate analysis (adjusted for the male sex, presence of constipation, Charlson Comorbidity Index (CCI) > 5, WBC > 15.25 × 10^9^/L, PCT > 0.09 ng/mL and blood urea nitrogen > 20 mg/dL) demonstrated that PCT > 0.09 (OR 4.38, 95% CI: 1.44–13.29, *p* = 0.009) and WBC > 15.25 × 10^9^/L (OR 3.27, 95% CI: 1.31–8.17, *p* = 0.011) were independently associated with major complications. PCT value of >0.09 ng/mL was identified as the best predictor for major complications with 84.8% (95% CI: 68.1–94.9) sensitivity and 51.8% (95% CI: 43.2–60.3) specificity in their study [33]. Additionally, there were significantly higher major complications for patients with PCT ≥ 1 ng/mL (32.4% vs. 15.3%, *p* = 0.019) [33]. The need for mechanical ventilation was reported only by Fransvea et al. and was required in eight patients (4.6%) [33]. However, the difference in the need for mechanical ventilation between patients with PCT < 1 ng/mL and PCT ≥ 1 ng/mL at admission is not statistically significant (*p* = 0.678).

Progression to sepsis was reported only by Fransvea et al. [33], where 10 patients (5.7%) progressed to sepsis. This was associated with higher PCT values at admission (0.21 ng/mL, 95% CI: 0.17–3.75 vs. 0.10 ng/mL, 95% CI: 0.05–0.53, *p* = 0.014). However, after multivariate analysis, only obstructive jaundice was independently associated with sepsis (Hazard ratio (HR): 5.46, 95% CI: 1.10–26.90, *p* = 0.038), while PCT was not significant (HR: 0.79, 95% CI: 0.19–3.31, *p* = 0.791). Additionally, patients with PCT < 1 ng/mL and PCT ≥ 1 ng/mL at admission had comparable progression to sepsis (n = 6 (4.4%) vs. n = 4 (10.8%), *p* = 0.136) [33].

LOS was reported by two authors [33,34]: there were 269 patients with pooled mean LOS of 8.53 ± 5.36 days. The difference in LOS between patients with PCT < 1 ng/mL and PCT ≥ 1 ng/mL at admission was non-significant (*p* = 0.126). Fransvea et al. reported overall mortality of 0.7% (n = 3) without difference for PCT < 1 ng/mL and PCT ≥ 1 ng/mL (*p* = 0.115) [33].

## 4. Discussion

PCT is produced by parafollicular C cells of the thyroid gland, and serum values are normally undetectable. A rise in PCT levels correlates with sepsis. However, evidence is scarce on its utility in diagnosing and prognosticating AC. AC is a common surgical admission that may result in morbidity and mortality. While the latest TG18 clearly describes the criteria for diagnosis and severity stratification of AC, the use of PCT as a prognostic marker needs to be better defined due to the lack of evidence. This is the first systematic review to evaluate the use of PCT in the management of AC patients. We demonstrated that the use of PCT is equivocal in predicting the severity of AC and clinical outcomes. However, PCT may be useful in predicting intra-operative difficulty.

AC is a common surgical admission and cholecystectomy is considered the “bread and butter” operation for general surgeons. AC requires timely diagnosis and risk stratification to prevent severe complications such as sepsis, multiple organ dysfunction syndromes (MODS), and multiorgan failure (MOF) [40,41,42,43,44,45]. TG18 is widely accepted in clinical care for AC [15]. The TG18/TG13 grades AC into Grade 1 (mild), Grade 2 (moderate), and Grade 3 (severe) [1]. Currently, leukocytosis and raised CRP levels are used clinically to diagnose, stratify severity and predict clinical progress and operative difficulty in AC [11,12,13,14,15]. Duration of symptoms has been correlated with the degree of inflammation and difficulty in safely identifying the Calot’s triangle [10]. Early presentation, prompt diagnosis, and index admission surgery are paramount in improving the clinical outcomes of AC patients. Embracing this philosophy, the traditional criterion of offering early index admission cholecystectomy < 72 h from symptom onset has been relaxed and extended to <1 week as a meta-analysis has shown that LC < 1 week from symptom onset also reduces complication rates compared to delayed interval LC > 4 weeks [46]. The included studies in our review however did not evaluate the role of PCT in predicting outcomes in index admission LC vs. delayed interval LC. This should be further evaluated.

While PCT has been shown to have higher specificity and sensitivity compared to CRP in the diagnosis of bacterial infections [32], a meta-analysis in 2007 on 18 studies assessed the use of PCT in critically ill patients and showed significant publication bias and heterogeneity of included studies [47]. Studies with smaller sample sizes were more likely to overestimate the diagnostic performance while the largest study had a diagnostic odds ratio of 1.94 and included the null effect [48]. The role of PCT in diagnosis and severity stratification was discussed in TG18 but its value was unable to be assessed due to a lack of evidence. Naz et al. [39] showed that a PCT cut-off of 1.65 ng/mL can distinguish Grade 2 from Grade 1 and 3 AC, and Yuzbasioglu et al. [25] showed that PCT ≤ 0.52 ng/mL can differentiate Grade 1 from Grade 2 or 3 AC. The extent of the rise in PCT is assumed to be correlated with the severity of sepsis. The study by Naz et al. [39] failed to show the use of PCT in discriminating Grade 3 from Grade 1–2 AC; however, a cut-off of 11.5 ng/mL was used (*p* = 0.535) and there was no description on the method used to derive the cut-offs. It is possible that a lower cut-off may be able to predict Grade 3 AC.

WBC is a marker of inflammation, and the degree of leukocytosis has been reported to correlate with severity. However, WBC count may be falsely lowered in certain groups of patients (e.g., old, diabetic, and immunosuppressed [16]. Leukopenia, on the other hand, while part of the systemic inflammatory response syndrome (SIRS) criteria (WBC < 4000/μL or >12,000 μL, or immature bands > 10%) [44], may not largely affect management. SIRS is less specific than the qSOFA score in predicting clinical outcomes of patients with acute hepatobiliary sepsis, with WBC contributing to the non-specificity [44]. These patients may be grouped as Grade 1 AC, for which index admission LC would be offered to surgically fit patients. In a single-center retrospective study including 149 patients with AC, Amirthalingam, V. et al. reported that the TG13 was too restrictive in guiding the management of AC patients [10]. The authors reported managing 98.8% (n = 83/84) Grade 1 AC and 90.8% Grade 2/3 AC (n = 59/65) patients with index admission LC [10]. American Society of Anaesthesiologists (ASA) score and co-morbidity scoring with CCI were deemed imperative to clinical decision-making and were introduced into the TG18 management algorithm for decisions for LC [15]. Thus, the use of WBC and PCT as inflammatory markers for AC diagnosis and severity grading should undergo further evaluation.

Intra-operatively, Wu et al. [36] showed that patients with PCT > 1.50 ng/mL were about 5 times more likely to have DLC than NDLC. They reported that patients with DLC were more likely to have dense adhesions, higher blood loss, longer operating time, and higher morbidity. In severe AC, surgery is more difficult due to increased inflammation which presents as edema and adhesions in the Calot’s triangle [25,46,49]. The severity of AC is an independent predictor of LOS and open conversion [50] and is also associated with resorting to bail-out strategies such as subtotal cholecystectomy, and higher 30-day mortality [51]. The increased open conversion rates may be due to dense adhesions caused by tissue inflammation and fibrosis of Calot’s triangle resulting in difficulty identifying critical structures, obtaining a critical view of safety, failure to progress, or surgeons’ self-determined threshold for open conversion i.e., “the inflection point” [52]. The changes to local anatomy predispose the patient to hemorrhage from the gallbladder bed or cystic artery and cause an increased risk of gallbladder perforation, stone spillage, and technical difficulties [53]. While other scoring systems, such as the G10 score [54], predict operative difficulty and risk of open conversion, it can only be completed intra-operatively (gallbladder appearance and presence of distension) and its use remain limited in the pre-operative phase. This warrants the importance of this review to identify biomarkers to predict severity and prognosticate outcomes in AC.

In our review, Fransvea et al. who studied 174 patients with AC showed that PCT > 0.09 ng/mL had high sensitivity (84.8%) in predicting major complications (overall incidence of death, need for mechanical ventilation, and conversion to open surgery) [33]. However, PCT did not significantly predict the use of mechanical ventilation or death alone. This could be due to the remote risk of post-operative mortality (0.1%–0.7%) and low-risk mechanical ventilation in AC patients [55]. Additionally, raised PCT level likely predicted the severity of AC and difficulty of surgery, which is correlated with worse clinical outcomes, rather than PCT directly predicting worse clinical outcomes. Hence, PCT may be used to determine if patients with severe AC should be preferentially treated with percutaneous drainage to achieve physiologic restoration before a definitive LC [15,33]. Yeo et al. have reported that percutaneous cholecystostomy restores the physiology within 48 h and patients with low CCI and suitable co-morbidity can have eventual definitive LC [56]. Though PCT can be used to aid clinical diagnosis of severe AC, the decision for percutaneous cholecystectomy should be based on other aspects including co-morbidity and physiological response to resuscitation.

Recently, Chan et al. emphasized the importance of proactive hepatobiliary consults in patients with DLC [57]. PCT levels could be used as a guide to consider pre-emptive hepatobiliary specialist consults by general surgical teams that routinely perform LC. Furthermore, in patients with DLC, a surgeon often resorts to bail-out strategies such as laparoscopic subtotal cholecystectomy before an open conversion. Though laparoscopic subtotal cholecystectomy reduces the risk of bile duct injury, it increases the risk of retained bile duct stones, bile leaks, and the need for secondary procedures such as endoscopic retrograde cholangiopancreatography [52]. This information on PCT will value-add to risk assessment, resource allocation, and patient education on disease progression and peri-operative risks. Adequate counseling of patients and their family members on peri-operative risks is required. Chia et al. showed that peri-operative consent was not truly “informed”, with only 44.4% of patients who could recall the serious complications of elective LC [58]. The role of PCT in predicting a bail-out procedure in patients with DLC is attractive and future studies could consider addressing this. PCT may also guide management and patient and/or family counseling on disease and post-operative risks.

PCT is also more effective than CRP and WBC in predicting sepsis [47,59,60,61], with a mean sensitivity of 0.77 (95% CI 0.72–0.81) and specificity of 0.79 (95% CI 0.74–0.84) [60]. In other inflammatory conditions such as acute pancreatitis, appendicitis, and cholangitis [62,63,64,65,66,67], PCT was more useful in severity stratification than CRP. In pancreatitis, PCT was found to predict sepsis (sensitivity: 86%, specificity: 96%) and multi-organ failure (sensitivity: 86%, specificity: 95%) [62]. In appendicitis, a PCT level of >0.5 ng/mL identified perforation or gangrene with 73% sensitivity and 94% specificity [64]. Moreover, PCT has a greater diagnostic value in identifying complicated appendicitis [68,69,70]. In diverticulitis, PCT was able to differentiate complicated cases of diverticulitis when combined with abdominal CT scans [71]. PCT has the potential to serve as a guide for antimicrobial therapy. Antimicrobial treatment is routinely used in AC patients but there is variability in clinical practice about the optimal duration of antibiotics [72,73]. In our institution, the use of antibiotics is limited to a short period of <5 days unless the patient has co-morbidities or is in septic shock and source control is not achieved. CRP has been demonstrated to be a useful biomarker to guide clinical response to antimicrobial and/or percutaneous therapy [74]. A CRP ratio (defined as the CRP value obtained at that particular week compared to the CRP value at week 1 of diagnosis) of ≤0.278 at week 3 was shown to be a good marker (sensitivity 0.786; specificity 0.714) for predicting antibiotic therapy of <5 weeks in pyogenic liver abscess [75]. PCT was also reported to guide the duration of antimicrobials for intra-abdominal infections [76,77,78,79]. However, cost-effectiveness remains to be determined, as PCT is considerably more expensive than CRP [80]. Our review could not determine whether PCT may serve as a guide to deciding antibiotic duration in AC as none of the included studies assessed this. Future prospective studies could study this issue.

The evidence from this systematic review shows the controversial utility of PCT to discriminate between Grades of AC, and more data is necessary to define the utility clearly. Unlike other acute sepsis pathologies, PCT lacks specificity in severity stratification. Furthermore, the cut-off values for diagnosis and severity stratification also remain to be determined. PCT cut-offs used in the included studies were heterogeneous and patient co-morbidities were also not included which could have affected outcomes [25,36,39]. In a prospective study including 124 patients with acute hepatobiliary sepsis (n = 83 patients with AC); Mak MHW et al. reported that diabetes mellitus predicted admission to the surgical high-dependency unit, myocardial infarction history predicted intensive care unit admission, hypertension, and hyperlipidemia predicted LOS, history of malignancy predicted morbidity [44]. Furthermore, the limitation of study heterogeneity was similarly shown in investigating the role of PCT in lower respiratory tract infections [81].

This study has its strengths. To our knowledge, this is the first study to summarize the literature regarding the use of PCT in the diagnosis, severity stratification, and prediction of postoperative complications in AC which may guide the management of AC. This is especially important given the lack of evidence on the use of PCT; TG18 also stated that the use of PCT on diagnosis and classification of severity is a question that should be addressed in the future [13]. There are, however, limitations to this study. We were unable to perform a quantitative synthesis of the results due to the small sample size, population heterogeneity, varied cut-off values, and reported complications. A range of 0.09 to 1.65 ng/mL for PCT was used in various studies to predict different clinical outcomes [33,39]. The methods used in determining these cut-off values were also heterogenous, making direct comparisons difficult. None of the included studies assessed PCT’s role in predicting antibiotic use duration. While most of the studies were prospective in nature, these were non-randomized studies, and the risk of bias assessment showed that the majority were only of moderate quality. There was no standardization of the timing at which PCT was measured, hence further studies can consider including the standardization of this. There was also no standardized definition of DLC [33,39]. While operating time will be increased in DLC, operating time is confounded by other factors, such as surgeon experience [36,82]. Lastly, this study is unable to confirm if adding PCT to existing institutional protocols of AC management could enhance care delivery.

## 5. Conclusions

This review demonstrated that PCT could predict severity, DLC, open conversion, and post-operative morbidity in AC in some studies. However, routine PCT use for the above could not be recommended until further well-designed studies address this knowledge gap. More prospective studies are necessary to define the role of PCT for it to be included in the next revision of guidelines.

## Figures and Tables

**Figure 1 medicina-59-00805-f001:**
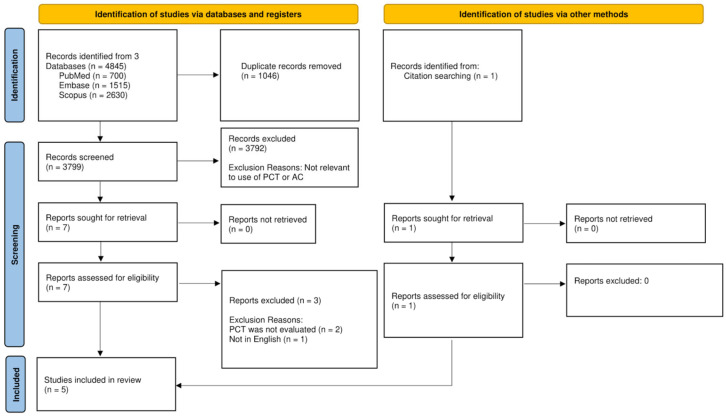
PRISMA Flowchart.

**Table 1 medicina-59-00805-t001:** Characteristics of included studies (n = 5).

Study, Year	Country	Study Period	Study Design	*n*	Age, Years	Sex, Male (M in %)	Co-Morbidities, n (%)	Diagnostic Criteria for AC	PCT Cut-Off Value (ng/mL)	Index Admission Cholecystectomy, n (%)	CRP, mg/dL	WBC, 10^9^/L	Outcome Variables Assessed
Fransvea et al., 2021	Italy	January 2015–December 2019	Retrospective cohort	174	62.33 ± 17.94	89 (51.1)	CCI: 2.33 ± 2.24 CHF: 7 (5.1%) CKD: 5 (2.9%) DM: 19 (10.9%) IHD: 11 (6.3%) PVD: 13 (7.5%)	TG18	≥1	174 (100)	25.00 ± 21.68	10.80 ± 4.93	Major Complication Conversion to open surgery Progression to sepsis Mechanical Ventilation Death Length of Stay
Sakalar et al., 2019	Turkey	June 2013–September 2014	Prospective cohort	95	59.87 ± 1.96	48 (50.5)	-	TG13	-	-	4.05 ± 6.07	11.00 ± 4.97	PCT Correlation with severity Operative Time Length of Stay
Wu et al., 2016	China	January 2017–April 2018	Retrospective cohort	115	53.54 ± 14.31	68 (59.1)	CAD: 4 (3.5%) COPD: 1 (0.9%) DM: 25 (21.7%) HTN: 26 (22.6%) Liver Cirrhosis: 14 (12.2%)	TG18	-	115 (100)	107.32 ± 39.71	12.54 ± 2.53	PCT Correlation with DLC Operative Time Blood loss Conversion to open surgery Dense adhesions Clavien-Dindo Post-op Grade Perforation of gallbladder
Yuzbasioglu et al., 2016	Turkey	July 2009–January 2011	Prospective cohort	200	59.97 ± 18.60	66 (33.0)	-	TG13	-	-	1.00 ± 2.30	10.40 ± 3.85	PCT Correlation with severity
Naz et al., 2021	India		Prospective cohort	104	-	-	-	TG18	-	-	-	12.05	PCT Correlation with severity

AC: Acute Cholecystitis; CAD: Coronary Artery Disease; CCI: Charlson Comorbidity Index; CHF: Congestive Heart Failure; CKD: Chronic Kidney Disease; COPD: Chronic Obstructive Pulmonary Disease; CRP: C-Reactive Protein; DLC: Difficult Laparoscopic Cholecystectomy; DM: Diabetes Mellitus; HTN: Hypertension; IHD: Ischemic Heart Disease; PVD: Peripheral Vascular Disease; TG: Tokyo Guidelines; WBC: White Blood Cell. All categorical variables are expressed as n (%), and all continuous variables are expressed as mean ± standard deviation unless otherwise specified.

**Table 2 medicina-59-00805-t002:** Relationship between procalcitonin (PCT) and severity, intra-operative and post-operative outcomes in acute cholecystitis (AC).

Study, Year	Clinical Severity of AC	Post-Operative Outcomes
Fransvea et al., 2021	PCT < 1 ng/mL vs. ≥1 ng/mL: Fever: 51 (37.2%) vs. 27 (73%), *p* < 0.001 Diffused peritonitis: 2 (1.5%) vs. 4 (10.8%), *p* = 0.019 Creatinine: 0.91 ± 0.34 vs. 1.12 ± 0.52, *p* = 0.015 BUN (mg/dL): 17.00 ± 8.99 vs. 25.00 ± 15.42, *p* < 0.001 Fibrinogen (mg/dL): 516.00 ± 233.76 vs. 691.00 ± 239.85, *p* = 0.002 CRP (mg/dL): 23.00 ± 20.23 vs. 32.00 ± 26.22, *p* = 0.013	PCT < 1 ng/mL vs. ≥1 ng/mL: Death: 1 (0.7%) vs. 2 (5.4%), *p* = 0.115 Mechanical Ventilation: 6 (4.4%) vs. 4 (10.8%), *p* = 0.678 Conversion to Open Surgery: 20 (14.6%) vs. 12 (32.4%), *p* = 0.013 Length of Stay (days): 8.77 ± 5.62 vs. 11.50 ± 9.18, *p* = 0.126 Progression to Sepsis: 6 (4.4%) vs. 4 (10.8%), *p* = 0.136 Major complications: 21 (15.3%) vs. 12 (32.4%), *p* = 0.019 PCT was not a significant factor in predicting progression to sepsis at multivariate analysis (HR: 0.79, 95%: 0.19–3.31, *p* = 0.791) Occurrence of Major Complications PCT > 0.09 ng/mL at admission had sensitivity (84.8%, 95% CI: 68.1–94.9) and specificity (51.8%, 95% CI: 43.2%–60.3%) for predicting the occurrence of major complications
Sakalar et al., 2019	PCT values had a correlation with severity of AC (*p* < 0.001)	PCT values had a non-significant correlation to length of hospital stay (*p* = 0.067)
Wu et al., 2016	Predicting DLC: PCT > 1.50ng/mL had sensitivity (91.3%, 95% CI: 78.3%–97.1%) and specificity (76.8%, 95% CI: 64.8%–85.8%) for predicting DLC The risk of DLC was significantly higher for patients with PCT > 1.50 ng/mL (OR: 5.2, 95% CI, 3.7–7.5, *p* = 0.004) compared to those with PCT < 1.50 ng/mL	NDLC vs. DLC Clavien-Dindo Post-Op Grade 1–2: 2 (2.9%) vs. 7 (15.2%) Clavien-Dindo Post-Op Grade 3–5: 0 (0%) vs. 1 (2.2%) Perforation of Gallbladder: 0 (0%) vs. 4 (8.7%) Conversion to Open Surgery: 0 (0%) vs. 1 (0.87%)
Yuzbasioglu et al., 2016	Discriminating Grade 1 from Grade 2 and 3 AC Cut-off: ≤0.52 AUC: 0.721 ± 0.037 Sensitivity: 95.45% Specificity: 46.67% *p*-value: <0.001 Discriminating Grade 2 from Grade 1 and 3 AC Cut-off: >0.14 AUC: 0.575 ± 0.043 Sensitivity: 62.3% Specificity: 56.83% *p*-value: 0.085 Discriminating Grade 3 from Grade 1 and 2 AC Cut-off: >0.8 AUC: 0.813 ± 0.053 Sensitivity: 72.4% Specificity: 90.06% *p*-value: <0.001	-
Naz et al., 2021	Discriminating Grade 1 from Grade 2 and 3 AC Cut-off: 0.15 AUC: 0.439 (95% CI: 0.289–0.589) Sensitivity: 91.7% Specificity: 20.6% *p*-value: 0.430 Discriminating Grade 2 from Grade 1 and 3 AC Cut-off: 1.65 AUC: 0.804 (95% CI: 0.629–0.978) Sensitivity: 71.4% Specificity: 88.9% *p*-value: 0.004 Discriminating Grade 3 from Grade 1 and 2 AC Cut-off: 11.5 AUC: 0.692 (95% CI: 0.428–0.957) Sensitivity: 100% Specificity: 61.5% *p*-value: 0.535	-

AUC: Area Under Curve; BUN: Blood Urea Nitrogen; CI: Confidence Interval; CRP: C-Reactive Protein; DLC: Difficult Laparoscopic Cholecystectomy; HR: Hazard Ratio; NDLC: Non-difficult laparoscopic cholecystectomy; OR: Odds Ratio; PCT: Procalcitonin. All categorical variables are expressed as n (%), and all continuous variables are expressed as mean ± standard deviation unless otherwise specified.

## Data Availability

Not applicable as no new data were created.

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
