# Peer review of "Should Procalcitonin Be Included in Acute Cholecystitis Guidelines? A Systematic Review"

_medicina, 2023, doi:10.3390/medicina59040805_

Round 1

Reviewer 1 Report

minor error along with the writing:

- line 35: please present datas as percentage or "rude" numbers, if mixed is difficult to interpret

- line 70: Pubmed is not cited but it is present in figure 1. Clarify it please

- line 257: there is a "10" without brackets, is it a citation?

Author Response

Response to Reviewer 1 Comments

Point 1: line 35: please present datas as percentage or "rude" numbers, if mixed is difficult to interpret.

Response 1: We thank the reviewer for the comment. We have made amendments to improve clarity of the statement. The sentnece now reads: “More than 200,000 patients are diagnosed with AC in the USA annually[4–7].”

Point 2: line 70: Pubmed is not cited but it is present in figure 1. Clarify it please

Response 2: We thank the reviewer for the comment. We apologise for the error. PubMed was the database that we searched instead of medline. We have updated line 70 accordingly. The sentnece now reads: “PubMed, Embase, Scopus, Web of Science, and the Cochrane Library were searched from inception till 21st August 2022 for articles studying the role of PCT in AC. The search was restricted to titles and abstract for all databases.”

Point 3: line 257: there is a "10" without brackets, is it a citation?

Response 3: We thank the reviewer for the comment. The “10” in line 257 is meant to be a citation and has been updated accoridingly on the manuscript.

Reviewer 2 Report

“Should Procalcitonin be Included in the Next Revision of To- 2 kyo Guidelines? A Systematic Review of Procalcitonin in Acute 3 Cholecystitis”

Reviewer response.

This is an interesting study and the authors have collected a unique dataset using cutting edge methodology. The paper is well written and structured.

This paper is a valuable confirmation of the fact that PCT must be added in the flow-chart to assess the severity of cholecystectomy.

I have some questions to better clarify some aspects of the paper.

Did the authors have found similar articles for that stratify the complexity of surgery or the inflammation in others acute inflammatory disease (appendicitis, diverticulitis)? A comparison with others papers may help to understand the value of PCT in other surgical procedures. 

Author Response

Response to Reviewer 2 Comments

Point 1: This is an interesting study and the authors have collected a unique dataset using cutting edge methodology. The paper is well written and structured.

This paper is a valuable confirmation of the fact that PCT must be added in the flow-chart to assess the severity of cholecystectomy.

I have some questions to better clarify some aspects of the paper.

Did the authors have found similar articles for that stratify the complexity of surgery or the inflammation in others acute inflammatory disease (appendicitis, diverticulitis)? A comparison with others papers may help to understand the value of PCT in other surgical procedures.

Response 1: We thank the reviewer for the positive comments on our manuscript. We have added statements to compare the use of procalcitonin in other acute inflammatory diseases. The sentence reads: “Moreover, PCT has greater diagnostic value in identifying complicated appendicitis[68–70]. In diverticulitis, PCT was able to differentiate complicated cases of diverticulitis when combined with abdominal CT scans[71].”

Reviewer 3 Report

Dear Editor,

I thank you for giving me the opportunity to review the manuscript entitled 'Should Procalcitonin be Included in the Next Revision of Tokyo Guidelines? A Systematic Review of Procalcitonin in Acute Cholecystitis. The subject of the article is research which interests me as well. That's why I read the article with excitement. The topic research is well-designed in my opinion. 

The article is about the study of procalcitonin, which may be a follow-up criterion in the management of acute cholecystitis patients, and in this regard, it may be important in the emergency room management of critically ill patients. In this respect, it was a very useful and informative article for emergency medicine and general surgery.

Author Response

Response to Reviewer 2 Comments

Point 1: Dear Editor,

I thank you for giving me the opportunity to review the manuscript entitled 'Should Procalcitonin be Included in the Next Revision of Tokyo Guidelines? A Systematic Review of Procalcitonin in Acute Cholecystitis. The subject of the article is research which interests me as well. That's why I read the article with excitement. The topic research is well-designed in my opinion. 

The article is about the study of procalcitonin, which may be a follow-up criterion in the management of acute cholecystitis patients, and in this regard, it may be important in the emergency room management of critically ill patients. In this respect, it was a very useful and informative article for emergency medicine and general surgery.

Response 1: We thank the reviewer for the positive comments on our manuscript.
